# Kinetic Parameters of *Saccharomyces cerevisiae* Alcohols Production Using *Nepenthes mirabilis* Pod Digestive Fluids-Mixed Agro-Waste Hydrolysates

**Nkosikho Dlangamandla**, **Seteno K. O. Ntwampe ***, **Justine O. Angadam**, **Boredi S. Chidi** **and Maxwell Mewa-Ngongang**

Bioresource Engineering Research Group (*BioERG*), Faculty of Applied Sciences, Department of Biotechnology, Cape Peninsula University of Technology, P.O. Box 652, Cape Town 8000, South Africa; dlangamandlans@gmail.com (N.D.); omajustine@gmail.com (J.O.A.); boredi2002@gmail.com (B.S.C.); mewamaxwell@gmail.com (M.M.-N.)

**\*** Correspondence: NtwampeS@cput.ac.za; Tel.: +27-214-609-097

**Abstract:** In this study, microbial growth kinetics and modeling of alcohols production using *Saccharomyces cerevisiae* were evaluated using different hydrolysates in a single pot (batch) system. Mixed agro-waste hydrolysates from different pre-treatment methods, i.e., *N. mirabilis*/CP and HWP/DAP/CP, were used as the sole nutrient source in the fermentations used to produce the alcohols of interest. The maximum *Saccharomyces cerevisiae* concentration of 1.47 CFU/mL ($\times 10^{10}$) was observed with HWP/DAP/CP hydrolysates, with a relative difference of 21.1% when compared to the *N. mirabilis*/CP cultures; the product yield based on biomass generation was relatively (20.2%) higher for the *N. mirabilis*/CP cultures. For the total residual phenolic compounds (TRPCs) generation, a relative difference (24.6%) between *N. mirabilis*/CP and HWP/DAP/CP pre-treatment systems was observed, suggesting that *N. mirabilis*/CP generates lower inhibition by-products. This was further evidenced by the lowest substrate utilization rate ($3.3 \times 10^{-4}$ g/(L·h)) for the *N. mirabilis*/CP cultures while achieving relatively similar product formation rates to those observed for the HWP/DAP/CP. A better correlation ($R^2 = 0.94$) was obtained when predicting substrate utilization for the *N. mirabilis*/CP cultures. Generally, the pre-treatment of mixed agro-waste using *N. mirabilis*/CP seemed appropriate for producing hydrolysates which *Saccharomyces cerevisiae* can effectively use for alcohol production in the biorefinery industry.

**Keywords:** agro-waste; alcohols; microbial growth; *Nepenthes mirabilis*; *Saccharomyces cerevisiae*

## 1. Introduction

Fermentation is a well-established process used for the production of bioproducts (i.e., bioethanol, biobutanol, isobutanol, lactic acid, citric acid, etc.) from glucose and/or lignocellulosic biomass hydrolysates [1]. However, the utilization of lignocellulosic biomass (agro-waste) hydrolysates as a sole carbon source largely relies on extractable and fermentable constituents in the biomass, i.e., holocelluloses, which can be extracted using pre-treatment technologies involving physical, chemical, and enzymatic hydrolysis, subsequent to fermentation to produce products, including alcohols using commercial strains of *Saccharomyces cerevisiae* [2–5]. Comparatively, *S. cerevisiae* is the most commonly used yeast for the production of bioethanol at an industrial scale [2], using easily fermentable constituents in a broth.

The challenges associated with the fermenter performance for alcohol production is largely attributed to the hydrolysates' inhibition during fermentation [6]. Inhibitors can also be directly

associated with hydrolysis methods used to extract fermentable total reducible sugars (TRS) from lignocellulosic biomass [6], resulting in stunted cell growth of the fermenters and further leading to low bioproduct concentration and fermenter productivity. Common inhibitory compounds are classified into three groups: 1) phenolic compounds (determined in this study), 2) furan derivatives, and 3) weak organic acids [7–9], largely produced during lignocellulosic biomass hydrolysis among fermentable holocellulose constituents including galactose, mannose, and xylose [10], with cellulose predominantly producing glucose [11–15]. Overall, the suitability of the pre-treatment/hydrolysis methods used (biological, physical, and chemical methods) has not been developed to reduce the toxicity of the constituents in the resultant pre-treatment hydrolysate.

Recent studies have shown that chemical hydrolysis has the ability to reduce inhibitory by-products, when compared to biological hydrolysis, which positively influences the productivity and biomass generation during alcohol production [16]. For most studies, hydrolysis is solely performed using cellulases, but there are other enzyme cocktails that can also be effectively used to perform both the delignification and holocellulolysis of renewable resources such as lignocellulosic biomass, including agro-waste, without the use of synthetic chemical and high-energy processes. These enzyme cocktails include those found in the pods of *Nepenthes mirabilis,* which were found to be suitable for holocellulolysis because they contain β-glucosidase, xylanases, and carboxylesterase [17]. However, the fermenter performance in the hydrolysate recovered from such *N. mirabilis* pod digestive fluid hydrolysis must be compared to the hydrolysates of combined conventional hydrolysis methods, i.e., hot water, dilute acid, and cellulases. This is an evaluation which can be understood using kinetic model parameter evaluations.

For an effective performance parameter determination, suitable mathematical kinetic models and experimental designs used for assessing the impact of fermenter conditions, including hydrolysates, are required [18]. The output of the kinetic models can facilitate assessment of optimal conditions and system control efficiency, including media (hydrolysate) selection [19]. Previously, Monod, Moser, Tessier, Logistic, and Leudeking-Piret models were used to describe the microbial growth, substrate consumption, and product formation rates [18–20]. Therefore, they can be used to comparatively analyze hydrolysate suitability. However, the selection of these models depends on the required purpose of the individual studies.

The purpose of this study was to determine the microbial growth, substrate utilization, and the product formation kinetic parameters during the fermentation processes using hydrolysates of *N. mirabilis*/cellulase (*N. mirabilis*/CP) in comparison to those of hot water/dilute acid/cellulase (HWP/DAP/CP)-mixed agro-waste hydrolysis systems for fermentations facilitated by a commercial South African *S. cerevisiae* strain (VIN13).

## 2. Materials and Methods

### 2.1. Confirmatory Identification of the Commercial Yeast Used for Fermentation

The genomic DNA (gDNA) extraction was performed according to a protocol analogous to that described in Zymo Research Catalogue No. D6005. DNA was extracted from the 24 h YPD pure yeast culture using the ZR DNA Kit (Zymo Research, Catalogue No. D6005, Irvine, CA, USA). The ITS target region was amplified using One Taq Quick-Load 2× Master Mix (NEB, Catalogue No. M0486, Ipswich, UK), using primers ITS1-5′-TCCGTAGGTGAACCTGCGG-3′ and ITS2-5′-TCCTCCGCTTATTGATATGC-3′, with repeated sequencing using forward 27F-5′-AGAGTTTGATCMTGGCTCAG-3′ and reverse 1492R-5′-GGTTACCTTGTTACGACTT-3′ primers (Nimagen, Brilliant Dye Terminator Cycle Sequencing Kit V3.1, BRD3-100/1000) to ascertain the correctness in the identification of the isolate (Zymo Research, Catalogue No. D4001, Irvine, CA, USA). The PCR products (i.e., extracted fragments) were run on a gel, and extraction with the Zymoclean™ Gel DNA Recovery Kit was performed thereafter. PCR was conducted in 100 μL reactions, while 100 ng of gDNA was used [21]. The PCR conditions were set at 36 cycles of 98 °C denaturation

for 30 s, primer annealing at 60 °C for 20 s, and elongation at 72 °C for 60 s. PCR products were further gel extracted (Zymo Research, Zymo Clean™ Gel DNA Recover kit) and purified (Zymo Research, ZR DNA sequencing clean-up kit Catalogue No. D4050, Irvine, CA, USA) while the resultant extracts were sequenced (forward/reverse direction). Thereafter, analysis with the ABI PRISM 3500xl Genetic analyzer ensued. The PCR products were further purified using a Zymo Research, ZR-96 DNA Sequencing Clean-up kit (Catalogue No D6006, Irvine, CA, USA) and analyzed using a CLC main workbench. Thereafter, the sequences generated were compared with the available nucleotide sequences in the NCBI Genbank database (http://blast.ncbi.nlm.nih.gov/Blast.cgi) for confirmatory identification of the *S. cerevisiae* strain used, with an accession number of KT32652.1 being assigned [22]. The *S. cerevisiae* was donated by a commercial producer of the yeast in South Africa (Anchor Yeast, Johannesburg, South Africa).

### 2.2. Fermentation Inoculum Preparation and Yeast Cell Counts

The yeast (10 mg) was grown in a medium containing 100 mL of Yeast Extract-Peptone-Dextrose (YPD) broth (i.e., yeast extract, 10 g/L; peptone, 20 g/L; dextrose, 20 g/L, at pH 6.4) incubated for 24 h at 30 °C. The yeast was further cultured on a Potato Dextrose Agar at 30 °C for 48 h. Colonies were further streaked out onto other PDA Petri-dishes to ensure the purity of the yeast used for inoculum. The inoculum was prepared using pure freshly-grown yeast cultures, with numerous colonies being picked to inoculate the 5 mL YPD broth followed by incubation for 24 h at 30 °C. To further prepare the fermentors, 100 μL of the overnight (24 h) YPD-yeast cultures were inoculated into individual flasks containing 50 mL of the hydrolysate from different hydrolysis methods and incubated for 72 h at 30 °C in a shaking incubator (LABWIT- ZWY-240, Shanghai Zhicheng Analytical, Shanghai, China) at 120 rpm. All fermentations were done in triplicate.

The quantification of yeast cell counts was performed using MediXgraph CFU Scope v1.5 software in order to determine the colony forming units (CFU/mL) on agar plates. This software is used as a mobile application for rapid and semi-automated CFU determination in a controlled environment [23]. The Optical Density (OD) of the yeast cultures during fermentation was determined using a Jenway 7305 UV/V is spectrophotometer (Cole-Parmer, St Neots, UK) at 600 nm as a further method to ascertain the accuracy of the kinetic parameter. This method also detects unculturable/non-viable biomass. All cell counts and OD measurements were done in triplicate.

### 2.3. Fermentation Medium Preparation Using Hydrolysates

The fermentation medium using various hydrolysates from two different agro-waste pre-treatment regimes were prepared as follows: First, the dried and milled waste (24 h/80 °C, >45 μm to <100 μm) was pre-mixed using a 1:1 ratio, i.e., 25% (*w/w*) for each waste, i.e., 1 g *Citrus sinensis peel*, 1 g *Malus domestica peel*, 1 g *Zea mays* cob, and 1 g *Quercus robur* (oak) yard waste were used based on their regional availability in the Western Cape, South Africa. Furthermore, the mixed agro-waste was pre-treated using conventional methods. This included, for example, treatment with hot water and dilute (1%) sulphuric acid, followed by commercial cellulases within each batch fermenter, and subsequently with *N. mirabilis* pod digestive fluids followed by enzymatic hydrolysis using commercial cellulases (Sigma-Aldrich, Darnstadt, Germany), i.e., within a single pot (batch) system and without the removal of hydrolysates during each treatment cycle or stage.

For *N. mirabilis* fermentations, the dried mixed agro-waste (2 g) was slurried in sterile distilled water (sdH$_2$O, 200 mL) using Erlenmeyer flasks (250 mL, in triplicate) fitted with sampling syringes [24] at 30 °C in a shaking (120 rpm) incubator (LABWIT- ZWY-240, Shangai Zhicheng Analytical, Shanghai, China), to ensure homogenization for 48 h. The slurry was further pre-treated with *N. mirabilis* digestive fluids, whereby a volume (2 mL) of *N. mirabilis* pod digestive fluids were added into individual Erlenmeyer flasks subsequent to further homogenization (96 h), with a volume of 1200 μL (600 μL per gram mixed agro-waste) of commercial cellulases (25 U/L, Sigma-Aldrich, Darmstadt, Germany) being added to each sodium acetate buffered (pH 4.5) fermentation flask for the furtherance

of the pre-treatment regime, up to a total experimental time of 168 h [25]. Thereafter, the final pre-treatment hydrolysates samples were centrifuged (4000× *g*, for 15 min) for solid sedimentation and filter sterilized (0.22 μm, cellulose filters, Isopore™, Burlington, MA, USA) subsequent to storage at −20 °C for further fermentation purposes.

Similarly, for conventional methods, a single pot pre-treatment strategy was followed with the hot water pre-treatment being conducted by slurrying 2 g of agro-waste (prepared as previously) in 200 mL sdH$_2$O using Schott bottles, subsequent to autoclavation (121 °C) for 15 min [26]. Thereafter, sulphuric acid constituting a final concentration of 1% (*v/v*) was added into each Schott bottle, subsequent to autoclavation at 120 °C for 15 min [26,27]. Thereafter, cellulases hydrolysis ensued by adding 600 μL of commercial cellulases (25 U/L, Sigma-Aldrich, Darmstadt, Germany) per gram of mixed agro-waste used. However, prior to cellulases addition, the pH was adjusted to 4.5 using a sodium acetate buffer. Centrifugation at 4000× *g* for 5 min to recover agro-waste free hydrolysates was instituted with filter sterilization of the supernatant (0.22 μm, cameo cellulase™, Isopore™, Burlington, MA, USA). The final samples were stored at −20 °C for further fermentation purposes.

## 2.4. Fermentation Conditions

The fermentation process in this study was conducted using hydrolysates from different mixed agro-waste pre-treatment processes. The filtrates from each pre-treatment method were used directly as a fermentation medium without any chemical supplementation. The fermentation was performed at 30 °C in 250 mL flasks containing 50 mL of the hydrolysates from the pre-treatment methods. A 10% (*v/v*) inoculum was prepared as previously highlighted, with a minimum *S. cerevisiae* concentration of $1 \times 10^6$ CFU/mL being used for each flask. The experiments were performed for 72 h in a shaking incubator (120 rpm), while 3.5 mL of samples were withdrawn every 24 h followed by centrifugation at 4000× *g* (5 min) and filtration (cameo 0.22 μm sterile syringe filter) prior to analyses. During the process, 10 μL of the diluted samples were further cultured into PDA plates incubated for 48 h at 37 °C to determine the colony forming units (CFU/mL). Furthermore, 1.4 mL of the samples was used to analyze for TRS, total residual phenolic compounds (TRPCs), and the presence of alcohols, including other analyses related to computations associated with kinetic parameter determinations [28]. A residual volume (2 mL) of the withdrawn sample was recycled back into the same flasks. All sample residues were stored at −20 °C for further use.

## 2.5. Analytical Methods

### 2.5.1. Total Reducing Sugar and Phenolic Compound Quantification

The TRS and its consumption during the fermentation were analyzed using the Dinitrosalicylic acid (DNS) method [29]. The aliquots were filtered, and 1 mL of each sample was transferred to a glass test tube subsequent to dilution using 9 mL of sdH$_2$O. The assay mixture contained 1.5 mL of the diluted aliquots and 1500 μL DNS reagent in sterile 15 mL glass test tubes subsequent to heating (90 °C) for 10 min. The assay mixture was cooled to ambient temperature prior to the addition of 0.5 mL of 40% (*w/v*) sodium potassium tartrate solution. The absorbance was determined using a Jenway 7305 UV/Vis spectrophotometer (Cole-Parmer, St Neots, UK) at 575 nm [29]. A calibration curve was developed using glucose (100 to 1000 mg/L) as the standard.

In this study, the total residual phenolic contents in the *N. mirabilis*/CP and HWP/DAP/CP pre-treatment hydrolysates were determined using the Folin-Ciocalteu reagent [30]. A volume (100 μL) of the *N. mirabilis*/CP and/or HWP/DAP/CP hydrolysates was added to an assay mixture containing 1.5 mL and 250 μL of sterile distilled water and the Folin-Ciocalteu reagent, respectively. Thereafter, a volume (1 mL, 20% *w/v*) of sodium carbonate was added after 3 min. Furthermore, the assay mixture was homogeneously mixed and stored in darkness for 1 h before the absorbance was read at 650 nm using a Jenway 7305 UV/Vis spectrophotometer (Cole-Parmer, St Neots, UK). A calibration curve was generated with 2 to 10 mg/L of 1,2-dihydroxybenzene in sdH$_2$O for further determination of TRPCs concentration [7,31].

2.5.2. Alcohol Determination

The alcohol production was determined using a modified GC-MS method developed by Reference [32]. A gas chromatograph (6890N, Agilent technologies network) attached to a CTC Analytics PAL autosampler and coupled to an Agilent Technologies inert XL EI/CI Mass Selective Detector (MSD) (5975B, Agilent Technologies Inc., Palo Alto, CA, USA) operated in a full scan mode with the source and quad temperatures at 230 °C and 150 °C, respectively, was used to quantify the production of alcohols—ethanol, butanol, and 2-phenylethanol. The separation of the fermentation broth volatiles was performed on a polar STABILWAX (60 m, 0.25 mm ID, 0.25 μm film thickness) Zebron 7HG-G007-11 capillary column. Helium was used as the carrier gas at a flow rate of 2 mL/min with the injector temperature being maintained at 250 °C. The sample was injected in a splitless mode. The oven temperature was maintained at 35 °C for 10 min and ramped up to 240 °C at a rate of 15 °C/min. The transfer line temperature was maintained at 250 °C with the mass spectrometer operated under electron impact mode at ionization energy of 70 eV and at a scanning range of 35–500 m/z [32].

*2.6. Kinectic Parameter Determination*

2.6.1. Monod and Microbial Growth Kinetic Parameters

The Monod model (Equation (1)) is the well-known model used to describe the proliferation of organisms under nutrient-rich conditions [33]. In this study, the Monod model was used to investigate the microbial growth kinetic parameters for *S. cerevisiae* using agro-waste hydrolysates from a single pot pre-treatment system as the sole carbon source.

$$\mu = \frac{\mu_{max}S}{K_s + S} \tag{1}$$

where $\mu_{max}$ is the maximum specific growth rate (h$^{-1}$) for unspecified reducible sugars, and $K_S$ is the estimated half-saturation constant (g/L), while $S$ is the residual TRS concentration (g/L).

The saturation constant reported herein, $K_s$, illustrated the rapidity of the microbial proliferation and its ability to attain a maximum specific growth rate ($\mu_{max}$) with the reducible sugars being utilized. The Monod's model is well-known to be applicable when there is a minimal presence of inhibitors, i.e., in the hydrolysates used as fermentation medium and as metabolic by-products produced by the fermenter during the fermentation. Similarly, Equation (2) was used for quantifying the yeast growth rate during fermentation and does consider the total biomass concentration as a single component; it is based on the modified Malthus equation [28]. Therefore, to further quantify the microbial growth, the Malthus equation was used as follows:

$$r_x = uX \tag{2}$$

whereby the biomass concentration, ($X$), and its maximum, ($X_{max}$), is described in colony-forming units (CFU/mL), taking into consideration the inoculum size ($X_0$).

2.6.2. Modelling TRS Consumption for Simultaneous Biomass and Product Formation

Theoretically, the TRS consumption in the fermentations is primarily and directly proportional to the biomass generation and product formed, with $p\frac{dX}{dt} \gg qX$ (Equation (3)). Therefore, in this study, TRS consumption was evaluated using the Luedeking–Piret model (Equation (3)) [19], assuming that the product formation is directly liked to biomass generation and product formation.

$$-\frac{dS}{dt} = p\frac{dX}{dt} + qX \tag{3}$$

whereby $p = 1/Y_{x/s}$ (g/CFU), while $q$ is the product formation coefficient (1/h). Therefore, Equation (3) can be rearranged as shown in Equation (4):

$$-dS = pdX + q\int X(t)dt \tag{4}$$

For the overall TRS consumption and to determine the residual TRS concentration, with the substitution of Equation (2) in Equation (4), followed by the integration with the initial conditions of $t = 0$ and $S = S_0$, Equation (5) was developed.

$$S = S_0 - pX_0\left\{\frac{e^{\mu_{max}t}}{\left\{1 - \left(\frac{X_0}{X_{max}}\right)(1 - e^{\mu_{max}t})\right\}} - 1\right\} - q\frac{X_{max}}{\mu_{max}}ln\left\{1 - \left(\frac{X_{max}}{\mu_{max}}\right)(1 - e^{\mu_{max}t})\right\} \tag{5}$$

### 2.6.3. Product Formation Kinetic Parameter Determination

The productivity of a fermentation system can be quantified using a modified Luedeking–Piret model (Equation (6)), whereby the parameters can be directly evaluated in relation to the fermentation data generated in particular, bio-products of interest [28].

$$P(t) - P_0 - n\left(\frac{X_{max}}{\mu_{max}}\right)ln\left[1 - \left(\frac{X_0}{X_{max}}\right)(1.0 - e^{\mu_{max}t})\right] = m[X(t) - X_0] \tag{6}$$

whereby $P$ is the product concentration (area %), while $n$ (area % mL/CFU·h) and $m$ (area % mL/CFU) are associated with the Luedeking–Piret model constants.

### 2.7. Data Handling, Relative Differences, and other Kinetic Parameters

The experimental data and kinetic models were computed and analyzed using Microsoft Excel 2013, while for other model parameters, Microsoft Excel Solver®was used. Furthermore, the relative differences were determined (Equations (7) and (8)) to illustrate the significance of the differences observed for the performance of the hydrolysates of the *N. mirabilis*/CP and the HWP/DAP/CP agro-waste pre-treatment systems. For reporting, the minimal microbial concentration detectable limit used was $log_{10}$ (CFU/mL) = 2. Other evaluated kinetic parameters are listed in Equations (9)–(12), with the reference amount for both the absolute and relative differences being the *S. cerevisiae* (VIN13) fermentations using hydrolysates from HWP/DAP/CP-agro waste pre-treatment systems.

$$\text{Absolute difference } = \left|New\ amount^{N.\ mirabilis/CP} - Reference^{HWP/DAP/CP}\right| \tag{7}$$

$$\text{Relative difference} = \frac{\text{Absolute difference}}{Reference^{HWP/DAP/CP}} \times 100 \tag{8}$$

$$Y_{x/s} = \frac{dX}{dS} \text{ (Biomass yield based on substrate consumption)} \tag{9}$$

$$Y_{p/x} = \frac{dP}{dX} \text{ (Product yield based on biomass generated)} \tag{10}$$

$$r_s = \frac{dS}{dt} \text{ (Substrate utilization rate)} \tag{11}$$

$$r_p = \frac{dP}{dt} \text{ (Product formation rate)} \tag{12}$$

## 3. Results and Discussion

### 3.1. Microbial Growth Parameters Using Mixed Agro-Waste Pre-Treatment Hydrolysates

In this study, the kinetics of cellular growth, substrate utilization, and alcohol production were determined using a commercial *S. cerevisiae* strain in hydrolysates obtained from the pre-treatment of mixed agro-waste constituted using peels of *C. sinensis* and *M. domestica*, including cobs of *Z. mays* and yard waste from *Q. robur*. A source of the hydrolysates is the newly proposed *N. mirabilis*/cellulases pre-treatment method, which was compared to conventional HWP/DAP/CP methods for the pre-treatment of lignocellulosic biomass. The maximum *S. cerevisiae* (VIN13) growth, a fermenter chosen for its rapid fermentations at ambient temperature with a high TRS and alcohol

tolerance, was determined using cellular counts (see Figure 1 and Table 1). This indicated a maximum cellular concentration of $1.47 \times 10^{10}$ CFU/mL for the HWP/DAP/CP hydrolysates compared to $1.16 \times 10^{10}$ CFU/mL attained for *N. mirabilis*/CP, with a differentiation quantified as a relative difference of 21.1%. This was attributed to the highest TRS concentration attained during the pre-treatment using the HWP/DAP/CP. This method is not specific to holocellulose extraction, but it does degrade the lignin further than the more holocellulose-targeted *N. mirabilis*-based pre-treatment method. This suggests that the *S. cerevisiae* strain would have had an adequate substrate supply during the fermentation process. However, maximum alcohol production was obtained using *N. mirabilis*/CP hydrolysates, which illustrated the limited inhibition characteristic of the hydrolysates as compared to those of the HWP/DAP/CP system. This observation was confirmed by the highest total residual phenolic content (TRPCs), which were 4.26 and 5.65 mg/L for *N. mirabilis*/CP and HWP/DAP/CP hydrolysates, respectively. The highest inhibition effect observed for the HWP/DAP/CP was only 1.31 area % mL/CFU $(\times 10^{-10})$ at a relative difference of 20.2% to that observed for *N. mirabilis*/CP.

Overall, the impact of the TRPCs has been observed with stunted product formation in HWP/DAP/CP, whereby the formed product was lesser than that in which the hydrolysates of *N. mirabilis*/CP were used as the sole nutrient media source. Furthermore, TRS consumption has been observed with an increase in the progression of the fermentation cycle. This showed that the metabolism of the *S. cerevisiae* strain used remained intact, and thus continued product formation during the fermentation process. Theoretically, a high concentration in TRS results in a high volume of alcohols being produced [23], unless there are inhibitory compounds in the hydrolysates, in which case the fermenter will use most of the available TRS to counteract the effects of the inhibitors [34]. The initial TRS concentrations ($S_o$) were 0.311 and 3.22 g/L, with the residual TRS concentrations ($S$) of 0.075 and 0.439 g/L at the end of the *N. mirabilis*/CP and HWP/DAP/CP fermentations, respectively. This further confirmed the suitability of using HWP/DAP/CP methods for delignification-cellulolysis operations.

The consumption of TRS during the fermentation has been previously studied using lignocellulosic biomass hydrolysates as the primary substrates, while *S. cerevisiae* was used as the fermenter [18,23,35]. Successes were attributed to the effectiveness of the pre-treatment methods used. By proposing a new method of pre-treatment, the achieved hydrolysates must perform similarly to the conventional methods with added beneficial attributes, or they should outperform the established methods, as is the case with the *N. mirabilis*/CP pre-treatment proposed herein. Furthermore, a comparison between the alcohols produced using *N. mirabilis*/CP and HWP/DAP/CP hydrolysates resulted in the hydrolysates of *N. mirabilis*/CP pre-treatment methods achieving a higher level of alcohol production than hydrolysates from the HWP/DAP/CP pre-treatment system (Table 1), with a relative difference of 5.2%. The higher TRS concentration from the HWP/DAP/CP pre-treatment system, i.e., 3.22 g/L, did not show a significantly higher alcohol production.

The biomass formation yield ($Y_{X/S}$) based on the TRS consumption and product formation yield ($Y_{P/x}$) based on the biomass generated was observed to be 4.92 and 0.53 (CFU/g $\times 10^{-13}$), and 1.58 and 1.31 area % mL/CFU $\times 10^{-10}$ for *N. mirabilis*/CP and HWP/DAP/CP respectively, with the HWP/DAP/CP hydrolysates showing a relatively rudimentary biomass yield (Table 1) which translated into a relative difference of 830%. To overcome such a momentous challenge, the results obtained in this study showed that the higher alcohol production and reduction of inhibition by-products reported as TRPCs can be achieved when *N. mirabilis*/CP pre-treatment is used, with the resultant hydrolysates being suitable as a media source for fermentation in the biorefinery industry. Moreover, and as an alternative, the cell density ($OD_{600}$) was also analyzed as shown in Figure 1b, whereby the optical density was similar for both conventional and *N. mirabilis*/CP hydrolysates fermentation. However, this method quantifies the turbidity of the total biomass in the samples irrespective of its activity and/or cultivability, which ultimately can include non-viable biomass.

**Table 1.** Resultant kinetic parameters for *S. cerevisiae* (VIN13) fermentations using hydrolysates from *N. mirabilis*/CP and HWP/ DAP/CP-agro waste pre-treatment.

| Hydrolysates | $X_{max}$ ($\times 10^{10}$ CFU/mL) | $S_o/S$ (g/L) | $P$ (%Area) (Ethanol; Butanol; 2-Phenylethanol) | $Y_{x/s}$ (CFU/g $\times 10^{-13}$) | $Y_{p/x}$ (Area % mL/CFU $\times 10^{-10}$) | $\mu_{max}/\mu$ (h$^{-1}$) | $K_s$ (g/L) | TRPCs (mg/L) |
|---|---|---|---|---|---|---|---|---|
| *N. mirabilis*/CP | 1.16 | 0.311/0.075 | 1.83 (1.23; 0.23; 0.38) | 4.92 | 1.58 | 1.76/0.095 | 1.32 | 4.26 |
| **HWP/DAP/CP** | 1.47 | 3.22/0.439 | 1.93 (1.02; 0.53; 0.38) | 0.53 | 1.31 | 1.58/0.088 | 7.46 | 5.65 |
| **Absolute difference** | 0.31 | n/d | - | 4.39 | 0.3 | 0.2/0.01 | 6.1 | 1.4 |
| **Relative difference (%)** | 21.1 | n/d | - | 829.9 | 20.2 | 12/9.1 | 82.3 | 24.6 |

$X_{max}$—maximum cell concentration ($\times 10^9$ CFU/mL), $S_o$—initial substrate (TRS) concentration (g/L), $S$—residual substrate (TRS) concentration (g/L), $P$—alcohol production (% area GC-MS), TRPCs—total residual phenolic compounds (mg/L), $Y_{x/s}$ [$X_{max}/(S_o-S)$]—biomass yield based on substrate consumption (CFU/g $\times 10^{-3}$), Alcohols—ethanol, butanol, 2-phenylethanol.

a）

b）

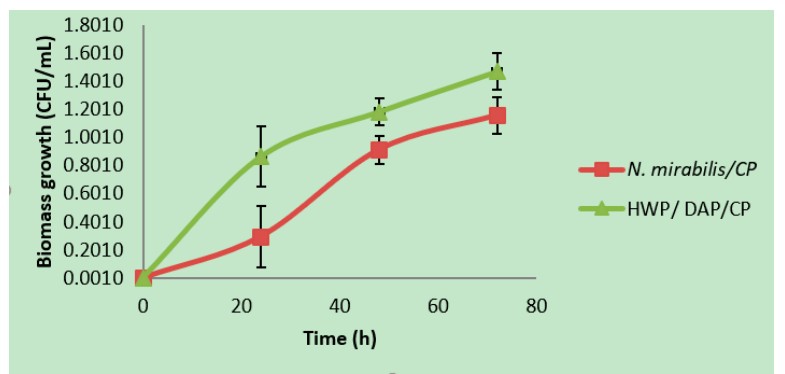
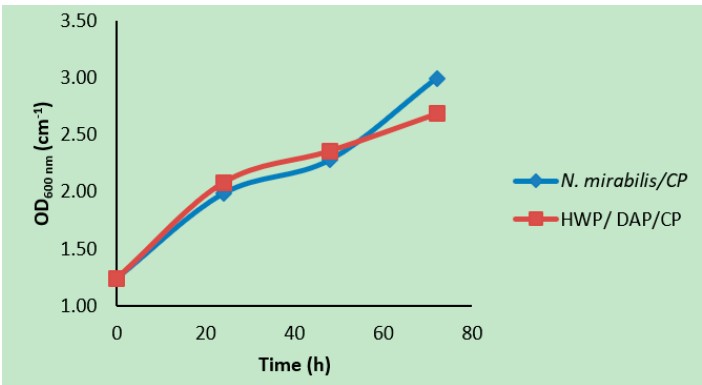

**Figure 1.** *S. cerevisiae* (VIN13) growth in *N. mirabilis*/CP and HWP/DAP/CP hydrolysates at 30 °C. (**a**) Biomass growth rate; (**b**) optical density.

### 3.2. Kinetic Rate Determinations

The kinetic rates of the *S. cerevisiae* facilitated fermentation were investigated during alcohol production. The Malthus model was used to determine and describe the microbial growth of the fermenter. The specific and maximum specific growth rates (Table 1) were 1.76 and 1.58 h$^{-1}$ (i.e., $\mu_{max}$) for *N. mirabilis*/CP and HWP/DAP/CP respectively, with the highest $\mu_{max}$ being observed for the *N. mirabilis*/CP hydrolysates, and with a significant relative difference of 12% and 9.1% for the specific growth rate. The $\mu_{max}$ value obtained in this study is relatively similar to what was reported in recent studies, whereby the *S. cerevisiae* was used as a fermenter with commonly used (refined) media [18]. However, in some studies, the $\mu_{max}$ related to *S. cerevisiae* in the batch system under acidic conditions was only 0.5717 h$^{-1}$ [2]. Furthermore, Table 2 illustrates additional kinetic parameters deemed important in this study, whereby a biomass formation rate was observed at 1.61 and $2.04 \times 10^8$ CFU/mL/h, with an absolute difference of 21.1% for *N. mirabilis*/CP and HWP/DAP/CP cultures, respectively. The product formation rate for both *N. mirabilis*/CP and HWP/DAP/CP was very low (0.025 and 0.027 area %/h, respectively) with a minute 5.26% relative difference considered to be insignificant at laboratory scale, but with significant product margin which was suggested to be large at an industrial scale. Similarly, substrate utilization rates of 0.0033 and 0.0387 g/(L·h) for *N. mirabilis*/CP and HWP/DAP/CP respectively indicated that a lot of the energy source (91.1% relative difference) was used for metabolic biomass maintenance functions other than product generation. This anomaly was previously attributed to high TRPCs in the hydrolysates of the HWP/DAP/CP pre-treatment regime used.

**Table 2.** Additional kinetic rates for *S. cerevisiae* (VIN13) fermentations using hydrolysates from *N. mirabilis*/CP and HWP/DAP/CP-agro waste pre-treatment.

| Parameter | Description (Units) | *N. mirabilis*/CP | HWP/DAP/CP | Relative Difference (%) |
|---|---|---|---|---|
| $r_x$ | Biomass formation rate ($\times 10^8$ CFU/mL/h) | 1.61 | 2.04 | 21.1 |
| $r_p$ | Product formation rate (area %/h) | 0.025 | 0.027 | 5.26 |
| $r_s$ | Substrate utilization rate (g/(L·h)) | 0.0033 | 0.0387 | 91.5 |

### 3.3. Product Formation Kinetics, Substrate Consumption Rate, and Modeling

Additional parameters related to product formation were quantified using the modified Luedeking–Piret model, which integrates TRS biomass concentration and product formation using *S. cerevisiae* in *N. mirabilis*/CP and HWP/DAP/CP hydrolysates. By further modifying the Luedeking–Piret model to fit the experimental data, Equation (13) was used by discarding the *n* constant function as in Equation (6), since it is a non-growth associated product formation term.

$$P(t) - P_0 = m[X(t) - X_0] \tag{13}$$

where a plot of $P(t) - P_0$ versus $[X(t) - X_0]$ would generate a linear trend-line with a slope *m* (area % mL /CFU), which illustrates the achievable rate of alcohol formation.

Table 3 illustrates the linkages of the product concentration with the cell density and Luedeking–Piret model constants. The achievable alcohol formation rate was observed at 1.0035 and 0.4848 area % mL/CFU in relation to the fermenter concentration. The correlation coefficients ($R^2$) of 0.941 and 0.4981 were obtained using modified Luedeking–Piret models for *N. mirabilis*/CP and HWP/DAP/CP hydrolysate cultures, respectively. However, the product formation for HWP/DAP/CP hydrolysates indicated a potentially non-related product formation to the generated biomass as observed with a lower correlation coefficient ($R^2$) of 0.481, which can be associated with inhibitors. Furthermore, the results obtained showed that the cell growth rate and alcohol production are interrelated with similar results by Reference [18], whereby the concentration of biomass increases resulted in the increase of the alcohols production, in particular for the *N. mirabilis*/CP hydrolysate

cultures. Therefore, with these results obtained, it was illustrated that the modified Luedeking–Piret model predictably demonstrated the product formation of the *N. mirabilis*/CP pre-treatment systems.

**Table 3.** Interlinkages of the product concentration with cell maximum density and substrate utilization as eluded by the Luedeking–Piret model for *N. mirabilis*/CP and HWP/DAP/CP hydrolysates.

| Parameter | Description (Units) | *N. Mirabilis*/CP | HWP/DAP/CP |
|:---:|:---:|:---:|:---:|
| $R^2$ | Correlation coefficient | 0.941 | 0.4981 |
| *m* | Slope (Area % mL/CFU) | 1.0035 | 0.4848 |
| *\*p* | g/CFU $\times 10^{12}$ | 2.0 | 18.9 |
| *\*q* | 1/h $\times 10^{-5}$ | 7.84 | 0.185 |

\* The Luedeking–Piret model constant value generated using Microsoft Excel Solver.

As the substrate was prepared by pre-treating mixed agro-waste to produce fermentable sugars, (total reducing sugar (TRS)), to produce alcohols using *S. cerevisiae,* Microsoft Excel solver was used to predict the rates constant (*p* and *q*) of the Luedeking–Piret model, as seen in Table 3. In Figure 2a,c, the response of the Luedeking–Piret model for substrate consumption was demonstrated, while Figure 2b,d show a comparison between the actual experimental and model-predicted data for substrate consumption. Substrate consumption theoretically illustrates the propensity of the microorganism to utilize the substrate for the purpose of producing the alcohols of interest, irrespective of the presence of inhibitory by-products from the pre-treatment of the biomass. When the Luedeking–Piret model was used for predicting the substrate utilization kinetic parameters, the coefficients determined for *p* were $2.18 \times 10^{-8}$ and $1.75 \times 10^{-7}$ (g/CFU), while for *q* the values attained were $7.84 \times 10^{-5}$ and $1.85 \times 10^{-4}$ (1/h) for the *N. mirabilis*/CP and HWP/DAP/CP fermentations, respectively (see Table 3). These values are related to the substrate consumption rate of the fermentations and the microorganism's ability to proliferate, which is assumed to be influential in product formation. The results obtained showed excellent correlation with the experimental data (with $R^2$ values of 0.9999 and 0.9999) for *N. mirabilis*/CP and HWP/DAP/CP, respectively, which confirmed the high significance of the model (see Figure 2). Similar results have been obtained by Reference [28], whereby the Luedeking–Piret model was used to describe the batch fermentation with an adequacy correlation coefficient ($R^2$) of 0.984 between the model and the experimental data.

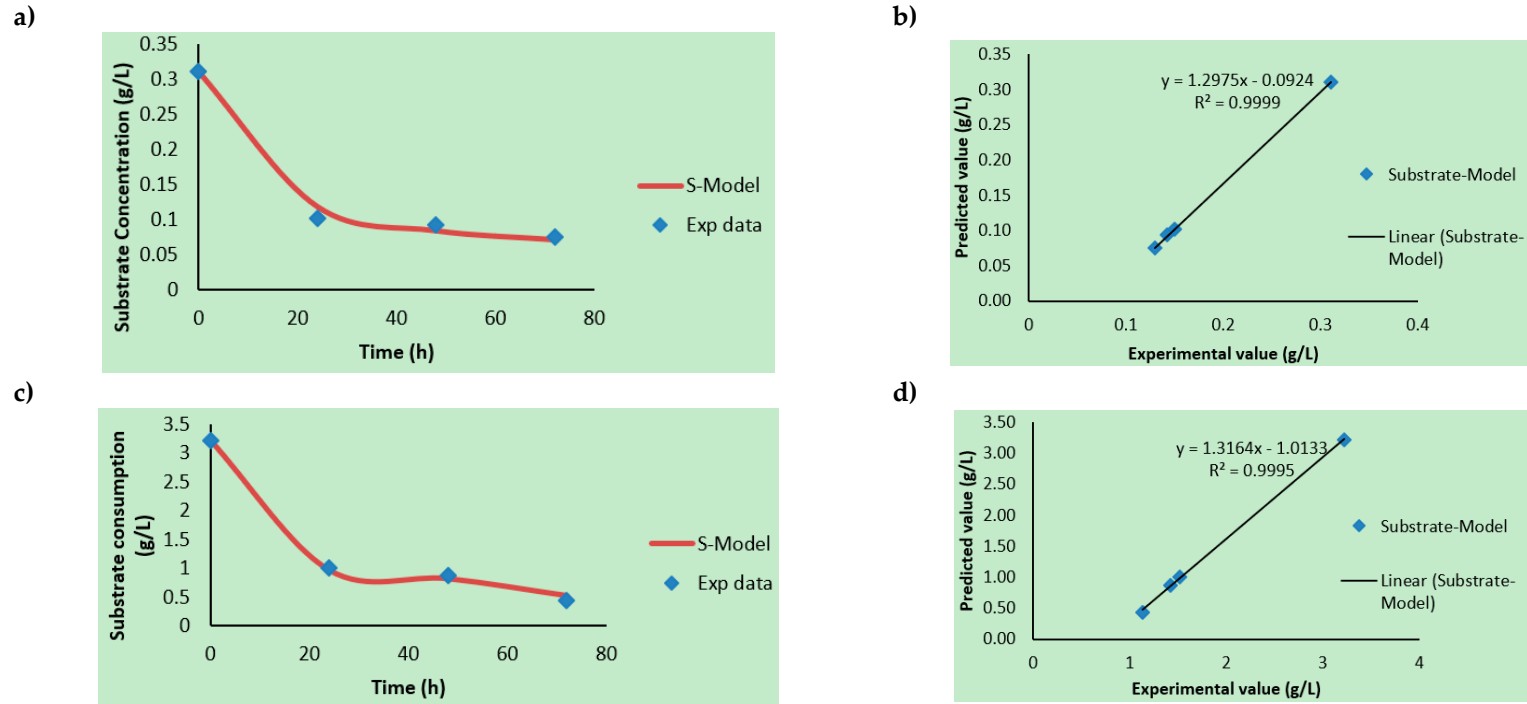

**Figure 2.** Prediction models: (**a**) The experimental data (dotted line) and model prediction (solid line) using *N.mirabilis*/CP; (**b**) a comparison of predicted value versus experimental data using *N.mirabilis*/CP filtrate as a substrate; (**c**) experimental data (dotted line) and model prediction (solid line) using hot water/dilute acid/cellulases; and (**d**) a comparison of the predicted value versus the experimental data using HWP/DAP/CP filtrate as a substrate.

## 4. Conclusions

In this study, kinetic parameters useful for the evaluation of cell growth rate, substrate consumption, and alcohol production for a single pot system under aerobic conditions in which *N. mirabilis*/CP and HWP/DAP/CP pre-treatment hydrolysates, were tested. Furthermore, this work demonstrated the feasibility of using the *N. mirabilis*/CP hydrolysates as the nutrient source to produce different alcohols, including ethanol, butanol, and 2-phenylethanol with *S. cerevisiae*. The proposed kinetic models showed that the *S. cerevisiae* maximum specific growth rate ($\mu_{max}$) for the *N. mirabilis*/CP and HWP/DAP/CP hydrolysates were 1.76 and 1.58 h$^{-1}$, respectively, showing that *N. mirabilis*/CP hydrolysates were a more effective nutrient source than those from the HWP/DAP/CP pre-treatment system for alcohol production. The maximum biomass concentration was found to be $1.47 \times 10^{10}$ CFU/mL/h with HWP/DAP/CP pre-treatment hydrolysates, which was related to the high TRS concentration obtained from the method, 3.22 g/L, which was reduced to 0.075 and 0.439 g/L for both *N. mirabilis*/CP and HWP/DAP/CP hydrolysate fermentation systems, respectively. Therefore, the substrate consumption during the process was found to not be directly proportional to the cell growth and alcohol formation, particularly for the HWP/DAP/CP hydrolysates fermentation systems. The amount of product formed was found to be lower for the HWP/DAP/CP hydrolysates fermentation. The suitability of the single pot multi-reaction system for applications in the biorefinery industry has also been demonstrated using the *N. mirabilis*/CP hydrolysates, hence the significant differences in the maximum cell biomass between *N. mirabilis*/CP and HWP/DAP/CP (conventional methods) pre-treatment hydrolysates in relation to the product formation. Additionally, the direct association between alcohols production and cell growth rates was further demonstrated using the modified Luedeking–Piret model. A better correlation coefficient ($R^2$) of predicted substrate utilization was observed for *N. mirabilis*/CP hydrolysates fermentation than for HWP/DAP/CP pre-treatment hydrolysates. Based on the results obtained, the pre-treatment of mixed agro-waste using *N. mirabilis/CP* proved useful in alcohol production using *S. cerevisiae*, a process which can be used for application in the biorefinery industry. Overall, the *N. mirabilis*/CP hydrolysates demonstrated a better fermentation yield than hydrolysates obtained using conventional methods, HWP/DAP/CP.

**Author Contributions:** Conceptualisation; J.O.A, N.D. and S.K.O.N. Draft preparations, methodology and investigation; J.O.A, N.D. and S.K.O.N. data interpretation, review, investigation and editing; B.S.C. and M.M.-N.

**Funding:** This research was funded by the Cape Peninsula University of Technology (CPUT) and University Research Fund (URF RK 16).

**Acknowledgments:** The authors would like to acknowledge funding from the Cape Peninsula University of Technology (CPUT) and University Research Fund (URF RK 16) for financing this project. The financial assistance of the South Africa National Research Foundation (NRF) towards this research is hereby acknowledged. We thank the Pan's Carnivores Plant Nursery for allowing us to use their plants' digestive fluids. We also acknowledge the technical support of the Bioresource Engineering Research Group (*BioERG*) team and biotechnology staff.

**Conflicts of Interest:** The authors declare no conflict of interest.

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
