# Peer review of "Kinetic Parameters of Saccharomyces cerevisiae Alcohols Production Using Nepenthes mirabilis Pod Digestive Fluids-Mixed Agro-Waste Hydrolysates"

_fermentation, doi:10.3390/fermentation5010010_

Round 1

Reviewer 1 Report

Dear Authors,

I reviewed your intriguing manuscript, which deals with mathematical modelling of two processes for yeast cultivation and alcohol fermentation based on hydrolysates of Nepenthes mirabilis pod extracts and cellulase-pretreated agro-waste, compared with traditional methods for agro-waste pretreatment. While traditional methods resulted in higher biomass formation, higher alcohol productivity was achieved by the enzymatic one-pot process. The developed mathematical models for growth and product formation were of high predictive power.

The work has considerable scientific merits, and might be a viable step towards enhanced biorefinery concepts. However, some improvement is needed to make the manuscript suitable for publication:

a) expression “bioethanol” (especially in title): This is misleading, because you describe the production of different alcohols (ethanol, butanol, phenylethyl alcohol, see line 191) by the fermentation processes. Suggestion to use “alcohols” (plural); “bio” is not needed in a scientific manuscript. By the way: line 191: why are ethanol and butanol produced by the yeast “bio”, and phenylethyl alcohol, also produced by the yeast, not?

b) Units: “g/L h” (line 23) should be “g/(L h)”. Check elsewhere in text (also in Tables)

c) 2.2: what was the pH-value for the liquid pre-culture (line 102-103)?

d) Some linguistic improvement /checking of typos is needed (e.g., line 150: “store” should be “stored”; line 301: “determined” should be “determine”)

e) line 152: “Numerous… n=2”: this is not really clear; is a number of 2 “numerous”?

f) line 204: µmax. is the “maximum specific growth rate”, not the “maximum growth rate”

g) line 214: “fermenter concentration (X)”: do you mean “biomass concentration”?????

h) line 246: “Z. mays” in italics

i) line 293: species names in italics, “(g/L)” for Ks not in italics

j) line 300: “Cerevisiae” should be “cerevisiae”

k) I do not find any info about which amounts of which alcohols was definitely produced, although you describe in 2.5.2 that you checked for ethanol, butanol, and phenylethyl alcohol.

Author Response

Response to Reviewer 1 Comments

I reviewed your intriguing manuscript, which deals with mathematical modelling of two processes for yeast cultivation and alcohol fermentation based on hydrolysates of Nepenthes mirabilis pod extracts and cellulase-pretreated agro-waste, compared with traditional methods for agro-waste pretreatment. While traditional methods resulted in higher biomass formation, higher alcohol productivity was achieved by the enzymatic one-pot process. The developed mathematical models for growth and product formation were of high predictive power.

The work has considerable scientific merits, and might be a viable step towards enhanced biorefinery concepts. However, some improvement is needed to make the manuscript suitable for publication:

Point 1:  Moderate English changes required 

Response 1: Spelling mistakes and English grammar has been checked. Extensive rewrite of sections has been done.

Point 2: Expression “bioethanol” (especially in title): This is misleading, because you describe the production of different alcohols (ethanol, butanol, phenylethyl alcohol, see line 191) by the fermentation processes. Suggestion to use “alcohols” (plural); “bio” is not needed in a scientific manuscript. By the way: line 191: why are ethanol and butanol produced by the yeast “bio”, and phenylethyl alcohol, also produced by the yeast, not?

Response 2: Thank you for this input. The use of bioethanol and biobutanol was just a mere and not so important emphasis on the employment of biological systems in the study.  Therefore, bioalcohols were change to alcohols throughout the manuscript.

Point 3:  Units: “g/L h” (line 23) should be “g/(L h)”. Check elsewhere in text (also in Tables)

Response 3: The correct units have been used throughout the manuscript.

Point 4:  2.2: what was the pH-value for the liquid pre-culture (line 102-103)?

Response 4:  pH-value of the pre-culture was added (line 106-107)

Point 5: Some linguistic improvement /checking of typos is needed (e.g., line 150: “store” should be “stored”; line 301: “determined” should be “determine”)

Response 5: Error (typos) were checked and corrected

Point 6: line 152: “Numerous… n=2”: this is not really clear; is a number of 2 “numerous”?

Response 6:  We agree, ‘n=2’ was confusing, and was removed. The ‘n’ initially referred to the number of pre-treatment methods. Not ‘numerous’ but ‘different’ rather.

Point 7: line 204: µmax. is the “maximum specific growth rate”, not the “maximum growth rate”

Response 7: The correct wording ‘specific’ was inserted.

Point 8: line 214: “fermenter concentration (X)”: do you mean “biomass concentration”?????

Response 8:  The correct wording ‘biomass’ was inserted.

Point 9: line 246: “Z. mays” in italics

Response 9: Thank you for this critical analysis, and for being attentive to details. Indeed, Z. mays has been corrected ( now in italics).

Point 10: line 293: species names in italics, “(g/L)” for Ks not in italics

Response 10:  Thank you for this critical analysis, and for being attentive to details. As requested, line 293 has been corrected.

Point 11: line 300: “Cerevisiae” should be “cerevisiae”

Response 11: Thank you for this critical analysis, and for being attentive to details. As requested, line 300 has been corrected.

Point 12: I do not find any info about which amounts of which alcohols was definitely produced, although you describe in 2.5.2 that you checked for ethanol, butanol, and phenylethyl alcohol.

Response 12:  The produced alcohols are now listed in Table 1. The concentrations are discussed under section 3.1.

Reviewer 2 Report

This is a nice manuscript, however, I have the following notes: 

The abstract looks like discussion, it must be and informative.

In materials and method, the gene of the yeast was presented but never referred to it in the discussion? 

The conclusion was not clear! For example how the finding would really help us to maximize the fermentation yield. 

Author Response

Response to Reviewer 2 Comments

This is a nice manuscript, however, I have the following notes: 

Point 1:  English language and style are fine/minor spell check required 

Response 1: Spelling mistakes and English grammar has been checked

Point 2: The abstract looks like discussion, it must be and informative.

Response 2:  The abstract has been improved. Please see track changes

Point 3:  In materials and method, the gene of the yeast was presented but never referred to it in the discussion? 

Response 3: The S. cerevisiae was donated by a commercial producer of the yeast in South Africa (Anchor Yeast, SA) and the analysis was for confirmatory purposes and re-identification of the commercial yeast used for fermentation.

The following was added as a part of the results and discussion. a fermenter chosen for its rapid fermentations at ambient temperature with a high TRS and alcohol tolerance – section 3.1.

Point 4: The conclusion was not clear! For example how the finding would really help us to maximize the fermentation yield. 

Response 4:  The conclusion section has been improved, by re writing a major section, with the following statement being included: The N. mirabilis/CP hydrolysates demonstrated a better fermentation yield than hydrolysates obtained using conventional methods, HWP/DAP/CP.